# Cardiac Hemangioma in the Right Atrium: Diagnostic Challenges, Imaging Clues, and a Novel Algorithm for Differential Diagnosis

**DOI:** 10.3390/life15121816

**Published:** 2025-11-27

**Authors:** Andrei Emanuel Grigorescu, Ramona Cristina Novaconi, Iulia Raluca Munteanu, Andrei Raul Manzur, Adrian Sebastian Zus, Mihai-Andrei Lazar, Oana Raluca Voinescu, Simina Crișan, Horea Bogdan Feier

**Affiliations:** 1Department VI Cardiology, “Victor Babes” University of Medicine and Pharmacy Timisoara, E. Murgu Sq. No. 2, 300041 Timisoara, Romania; grigorescu.andrei@umft.ro (A.E.G.);; 2Advanced Research Center, Institute for Cardiovascular Diseases, 300310 Timisoara, Romania; 3Institute for Cardiovascular Diseases of Timisoara, Clinic of Cardiovascular Surgery, Gheorghe Adam Street, No. 13A, 300310 Timisoara, Romania; 4Doctoral School Medicine-Pharmacy, “Victor Babes” University of Medicine and Pharmacy Timisoara, E. Murgu Sq. No. 2, 300041 Timisoara, Romania; 5Department of Pulmonology, Center for Research and Innovation in Precision Medicine of Respiratory Diseases, “Victor Babes” University of Medicine and Pharmacy Timisoara, Eftimie Murgu Sq. No. 2, 300041 Timisoara, Romania

**Keywords:** cardiac hemangioma, right atrium, echocardiography, cardiac MRI, coronary angiography, surgical resection

## Abstract

Background: Primary cardiac tumors are exceedingly rare, accounting for less than 0.2% of cases in autopsy series. Myxomas represent the majority, while hemangiomas are exceptionally uncommon, accounting for less than 2% of benign cardiac tumors. Their rarity, nonspecific clinical presentation, and overlapping imaging features make preoperative diagnosis particularly challenging. Case presentation: We report the case of a 47-year-old woman with exertional dyspnoea and a large right atrial mass initially suggestive of myxoma on transthoracic echocardiography. Coronary angiography revealed a small fistulous connection, raising suspicion of a vascular lesion. Surgical resection was performed, and histopathological examination confirmed a cavernous hemangioma. The postoperative course was favorable, with no residual mass on follow-up imaging. Discussion: Cardiac hemangiomas are often misdiagnosed as myxomas due to similar clinical and echocardiographic appearances. This case illustrates the diagnostic challenges posed by cardiac hemangiomas and the importance of correlating multiple imaging modalities when assessing atypical atrial masses. Our case highlights the potential role of multimodal imaging, including contrast echocardiography, cardiac MRI, and coronary angiography, in differentiating vascular from non-vascular cardiac tumors. Based on recurrent patterns in the literature, we outline a conceptual diagnostic framework that may assist preoperative evaluation. Conclusions: Although rare, cardiac hemangiomas should be considered in the differential diagnosis of atrial masses. Multimodal imaging may improve diagnostic accuracy, but additional data from multicenter registries are required to establish validated diagnostic pathways.

## 1. Introduction

Primary cardiac tumors are a rare clinical entity, with an incidence between 0.0017% and 0.19% in autopsy studies [1,2,3,4,5,6,7,8]. Benign tumors predominate, among which atrial myxomas represent 50–75% of cases [6,7,8]. In contrast, cardiac hemangiomas are exceptionally uncommon, accounting for less than 2% of benign cardiac tumors [1,2,3,5]. These vascular tumors are composed of proliferating endothelial cells forming abnormal vascular channels, and histologically, they are classified as cavernous, capillary, mixed, or arteriovenous. Cavernous hemangiomas are the most frequently reported subtype [4,9,10,11].

Although histologically benign, cardiac hemangiomas may exert clinically significant effects through their location and size. Depending on anatomical site, they can obstruct blood flow, interfere with valvular function, or provoke arrhythmias. Symptomatology is highly variable, ranging from incidental findings in asymptomatic patients to dyspnea, chest pain, palpitations, or even embolic events. The right atrium and right ventricle appear to be the most common sites of involvement, raising hypotheses about potential anatomical or hemodynamic predispositions [1,2,5,12,13,14,15,16,17].

Their low prevalence and heterogeneous clinical presentation contribute to the difficulty of establishing a reliable preoperative diagnosis. Conventional transthoracic echocardiography remains the first-line imaging modality, yet its capacity to distinguish between vascular and non-vascular tumors is limited. Recent reports suggest that contrast-enhanced echocardiography and cardiac magnetic resonance (CMR) can help distinguish vascular from non-vascular masses, while coronary angiography may provide additional clues when tumor vascularity or fistulous connections are suspected. However, no standardized diagnostic pathway currently exists, and most hemangiomas are identified only after surgical resection and histopathological examination [18,19,20,21,22,23,24,25,26,27].

The increasing number of cases reported in recent years likely reflects both improved imaging technologies and growing clinical awareness. Nonetheless, data synthesis highlights persistent gaps: lack of standardized classification, heterogeneity in follow-up protocols, and frequent preoperative misdiagnosis as myxoma [1,5,13,19,21,28,29].

In this context, we report the case of a 47-year-old woman with a right atrial cavernous hemangioma that was initially suspected to be a myxoma. Beyond the uncommon nature of this tumor, the case illustrates the limitations of routine diagnostic pathways and emphasizes the contribution of complementary imaging techniques in refining preoperative assessment. By integrating clinical findings with the most recent literature, we propose a structured diagnostic algorithm that may improve recognition of this underdiagnosed entity and support more accurate preoperative decision-making.

## 2. Case Presentation

A 47-year-old woman with significant cardiovascular risk factors—chronic smoking, class I obesity, mixed dyslipidemia, and essential hypertension—was referred to our center after progressive exertional dyspnea and fatigue corresponding to New York Heart Association (NYHA) class II. Her medical history was notable for total thyroidectomy in 2021, following levothyroxine therapy for iatrogenic hypothyroidism. The patient had no relevant family history of cardiovascular disease, and no psychosocial or genetic factors were identified as contributory.

On physical examination, the patient was afebrile, hemodynamically stable, and in sinus rhythm. Cardiac auscultation revealed a systolic murmur best heard at the lower left sternal border. No peripheral edema or signs of systemic congestion were present. Resting electrocardiography showed sinus tachycardia (102 bpm) with left axis deviation, while chest radiography was unremarkable. Routine laboratory investigations, including complete blood count and renal function, were within normal limits.

Diagnostic evaluation began with transthoracic echocardiography, which served as the initial modality for assessing the atrial mass.

Transthoracic echocardiography performed at admission demonstrated a large, echodense intracavitary mass occupying most of the right atrium, measuring 5.5 × 3.5 cm in maximal dimensions, suggestive of an attachment at the fossa ovalis, mimicking the typical location of atrial myxoma. Differential diagnoses considered at presentation included atrial myxoma, other benign atrial tumors such as lipoma or fibroelastoma, and less likely malignant cardiac tumors. The mass caused functional tricuspid stenosis, associated with moderate tricuspid regurgitation, and mild degenerative mitral regurgitation. Pulmonary artery systolic pressure was mildly elevated (35 mmHg). Left and right ventricular systolic function was preserved (LVEF 50–55%, TAPSE 2.5 cm) without pericardial effusion (Figure 1).

Coronary angiography showed a right-dominant coronary system without atherosclerotic stenoses. Incidentally, a small fistulous connection was observed, apparently draining into the right atrium or directly into the tumor mass, suggesting the presence of a hypervascular lesion. Despite this finding, the preoperative working diagnosis remained atrial myxoma (Figure 2) [5,18,19,20].

The transthoracic echocardiography and angiographic acquisitions presented in this report are subject to intrinsic resolution constraints, reflecting the clinical conditions under which they were obtained. The figures included represent the most illustrative frames, and additional optimization and annotation have been applied to improve clarity. Non-essential views have been removed to enhance the overall quality of the visual documentation.

Taken together, these findings suggested an atypical right atrial mass with possible vascular supply.

Surgical intervention was undertaken due to obstructive hemodynamic compromise and embolic risk. Median sternotomy and cardiopulmonary bypass with aortic cross-clamping were performed. Intraoperative inspection revealed a large sessile mass exceeding 5 cm, broadly attached to the interatrial septum. The mass was excised en bloc along with the affected septal tissue, and the resulting atrial septal defect was repaired using an autologous pericardial patch. The procedure was completed uneventfully, with cardiopulmonary bypass and ischemia times of 102 and 50 min, respectively. The patient was weaned from cardiopulmonary bypass in stable sinus rhythm (Figure 3).

Gross examination of the resected specimen showed a spongy, nodular tissue fragment measuring 5.5 × 5 × 3 cm, with a brownish, heterogeneous cut surface. Histopathological evaluation revealed a well-demarcated but non-encapsulated vascular lesion composed predominantly of medium-to-large dilated vascular channels, lined by a single layer of benign endothelial cells without atypia or mitotic activity. The stroma was variably fibrous and focally myxoid, with mild inflammatory infiltrates and adjacent myocardial hypertrophy. These features confirmed the diagnosis of cavernous hemangioma (Figure 4) [1,10,30,31,32].

The postoperative course was favorable. Early postoperative echocardiography demonstrated reduction of tricuspid regurgitation and preserved biventricular function. At discharge, the patient was in good general condition, hemodynamically stable, and showed no evidence of residual intracardiac mass or pericardial effusion. She was advised to continue antiplatelet therapy, thyroid hormone replacement, and general lifestyle modifications. Follow-up with cardiology was arranged at 1, 3, 6, and 12 months and annually thereafter. Antiplatelet therapy consisted of 75 mg of aspirin daily, and levothyroxine was maintained at a dose of 100 µg daily. No modifications of therapy were required during hospitalization or follow-up, and treatment was well tolerated. No adverse or unanticipated events were observed during hospitalization or follow-up.

The patient expressed satisfaction with the outcome of the intervention and reported significant improvement in symptoms, particularly exertional dyspnea and fatigue. She described the recovery period as uneventful and was reassured by the absence of residual mass on follow-up echocardiography. The patient emphasized the importance of early diagnosis and expressed appreciation for the comprehensive care received throughout the hospitalization and follow-up. Written informed consent was obtained from the patient to publish this case report and the accompanying images.

The clinical timeline of the patient is summarized in the following table (Table 1).

## 3. Discussion

Cardiac hemangiomas are exceedingly rare benign vascular tumors, representing less than 2% of benign primary cardiac tumors. Among reported subtypes, cavernous hemangiomas predominate, as confirmed in our case. Although histologically benign, their clinical impact is largely dictated by location and size, which can lead to obstruction, arrhythmias, embolic complications, or valvular dysfunction [1,2,3,5,20,28,33].

For completeness, the histopathological features of cardiac myxomas—often considered in the differential diagnosis—should also be recalled. Myxomas typically consist of stellate or polygonal cells embedded in a myxoid stroma, frequently accompanied by areas of hemorrhage, inflammatory infiltrates, and occasional calcifications. These tumors usually lack a true vascular architecture, which contrasts with the cavernous spaces lined by endothelial cells observed in cardiac hemangiomas. This histological distinction explains the overlapping clinical and echocardiographic appearance but underscores the importance of tissue examination for definitive diagnosis.

Our patient presented with progressive exertional dyspnea and fatigue, in line with the most frequently described symptoms in recent systematic reviews [1,12,13,16,34,35]. The intracavitary right atrial location is also consistent with published data, as approximately one quarter of cardiac hemangiomas arise in this chamber [2,14,15,20,29]. Importantly, the tumor’s broad attachment to the interatrial septum and echocardiographic appearance strongly mimicked atrial myxoma, a common diagnostic pitfall. This misclassification has been repeatedly reported in the literature, reflecting the overlapping features of these two entities [5,18,36,37,38].

### 3.1. Diagnostic Challenges and Imaging Clues

Echocardiography remains the first-line tool for detecting cardiac tumors, used in over 80% of reported hemangiomas. However, its ability to differentiate vascular from non-vascular masses is limited. In our case, echocardiography demonstrated a large echodense mass with hemodynamic impact but failed to suggest its vascular nature [1,21,23,39]. Coronary angiography, performed for preoperative assessment, incidentally revealed a small fistulous connection draining into the right atrium. While this finding retrospectively supports the diagnosis of a vascular tumor, it was not considered conclusive preoperatively. Similar observations have been reported in rare cases, where coronary angiography demonstrated tumor blush or anomalous vascular supply, offering indirect diagnostic clues [18,19,40,41].

Cardiac magnetic resonance (CMR) and contrast-enhanced echocardiography (CEUS) have been proposed as additional modalities that can reveal hypervascularity, T2-weighted hyperintensity, and persistent contrast uptake—features more suggestive of hemangioma than myxoma. Nevertheless, their use remains sporadic, and no standardized pathway has been established [18,21,22,23,24,25,26,39,40,42,43]. Moreover, access to advanced imaging modalities such as cardiac MRI and contrast-enhanced echocardiography is not uniform across cardiology centers, and their diagnostic criteria for cardiac hemangioma have not been validated in prospective studies. These methodological constraints, combined with the exceptional rarity of the tumor, inherently limit the generalizability of our observations. As a result, imaging-based distinctions between vascular and non-vascular cardiac masses should be interpreted with caution, and further multicenter experience will be required to determine their reproducibility and diagnostic accuracy.

Importantly, no formal sensitivity or specificity values have been established for CEUS or cardiac MRI in the diagnosis of cardiac hemangioma, as all available evidence derives from isolated case reports rather than systematic evaluations. Nonetheless, certain imaging patterns have been consistently associated with hemangiomas, including persistent intralesional contrast enhancement on CEUS and strong, homogeneous late gadolinium enhancement on cardiac MRI. These findings may increase diagnostic confidence, but they cannot replace histopathology in the absence of validated diagnostic performance metrics.

### 3.2. Proposed Diagnostic Algorithm

Drawing on our case and previously published evidence, we suggest a structured multimodal imaging algorithm for the evaluation of atrial masses suspected to be myxomas but exhibiting atypical features:Transthoracic echocardiography—initial detection and assessment of hemodynamic impact [1,5,21,24].Contrast echocardiography (CEUS)—aiming to assess vascularity; persistent contrast enhancement should raise suspicion for hemangioma [5,18,21,25,26,27,42].Cardiac MRI—aiming to refine tissue characterization; hyperintensity on T2-weighted imaging and strong post-contrast enhancement support a vascular lesion [22,23,25,26,27,39,40].Coronary angiography—aiming to evaluate tumor perfusion or identify coronary fistulas, particularly when surgical planning is required [19,20,26,27].

Although a stepwise diagnostic framework may support clinical reasoning in selected cases, no standardized or validated diagnostic protocol currently exists for cardiac hemangioma. The evidence base is limited to isolated case reports and small series, which precludes any meaningful quantitative comparison between our proposed framework and existing diagnostic pathways for cardiac masses in general. In this context, the scheme proposed here should be regarded as a conceptual aid rather than a guideline-level recommendation. Its structure reflects recurrent imaging patterns described in the literature and is informed by our prior systematic review, which identified all published cases over the past five years and confirmed the absence of any prospectively validated diagnostic approach for this rare tumor. Accordingly, the proposed framework is exploratory and hypothesis-generating, and its diagnostic utility will need to be assessed through larger case series or multicenter registries. Accordingly, the framework should be interpreted as exploratory and used to guide clinical reasoning in atypical cases, rather than as a prescriptive diagnostic algorithm. The proposed framework is exploratory and hypothesis-generating, and its diagnostic utility will need to be assessed through larger case series or multicenter registries.

While histopathological confirmation remains the gold standard, the adoption of such a multimodal approach could improve preoperative diagnostic accuracy and prevent misclassification (Figure 5).

### 3.3. Subdiagnosis and Implications

We hypothesize that cardiac hemangiomas are underdiagnosed. Many cases historically labeled as myxomas might have actually represented unrecognized hemangiomas, particularly in the absence of detailed histopathological examination or advanced imaging. The incidental angiographic finding in our patient illustrates how additional imaging modalities can reveal overlooked diagnostic clues. Given the increasing number of reports in recent years, it is plausible that this trend reflects not only improved detection but also correction of previous misclassification [1,2,11,28,29,33,37,38,39,44].

### 3.4. Surgical Management and Prognosis

Surgical resection remains the treatment of choice, performed in approximately 90% of published cases. The intervention is both therapeutic and diagnostic, alleviating symptoms, preventing complications, and providing tissue for histopathology. In our patient, complete resection of the mass and septal repair led to excellent short-term recovery. Literature data indicate low recurrence rates and favorable long-term outcomes following complete excision. Nevertheless, heterogeneity in follow-up practices remains a significant limitation, with most reports documenting only short-term surveillance [12,13,14,15,16,17,18,19,20,28,29,31,34,35,36,40,41,44,45,46,47,48,49,50,51,52,53,54,55,56,57,58,59,60,61,62,63,64,65,66,67,68]. The overall prognosis of cardiac hemangiomas is generally favorable following complete surgical excision, although data on long-term outcomes remain limited due to the rarity of the condition.

### 3.5. Limitations

Advanced imaging modalities such as cardiac MRI or contrast-enhanced echocardiography were not available in this case, which may have limited the refinement of preoperative tissue characterization and vascular assessment. Furthermore, this report describes a single instance of right atrial cavernous hemangioma, and its generalizability is therefore inherently restricted. As highlighted in our recent systematic review, only about 55 cases have been published in the past five years, underscoring both the exceptional rarity of this tumor and the absence of prospective data [5]. Under these circumstances, a quantitative evaluation or validation of the proposed diagnostic framework is not feasible; accordingly, the algorithm should be regarded as exploratory and hypothesis-generating. Larger case series and multi-center registries will be necessary to assess its diagnostic performance and potential clinical impact.

Given the absence of validated diagnostic pathways, the proposed framework should be interpreted strictly as a conceptual synthesis of current evidence rather than as a prescriptive clinical guideline.

Although cardiac MRI and contrast-enhanced echocardiography were not performed in this patient, their inclusion in the algorithm reflects their described diagnostic value in previously published cases rather than their use in the present work. As such, the framework incorporates modalities supported by the literature even when not available in this individual case.

### 3.6. Future Perspectives

This case highlights the current absence of standardized diagnostic or follow-up pathways for cardiac hemangiomas. Because available evidence is limited to isolated reports and small series, no validated algorithm exists for this entity. The conceptual framework we propose is therefore exploratory and informed not only by the present case but also by patterns identified in our prior systematic review, which synthesized all published cases over the past five years. Nevertheless, the lack of prospective validation limits its applicability, and caution is warranted when extrapolating its potential diagnostic value.

Another limitation lies in the intrinsic resolution constraints of the available echocardiographic and angiographic images, which reflect the technical conditions at the time of acquisition. Although optimized for clarity, their interpretability may remain imperfect.

## 4. Conclusions

Right atrial cavernous hemangiomas are rare and frequently mimic atrial myxomas on clinical and standard imaging grounds. Our case highlights the diagnostic pitfalls and reinforces the pivotal role of histopathological confirmation.

By integrating clinical findings with current literature, we emphasize two key conclusions:Cardiac hemangiomas are likely underdiagnosed, frequently misclassified as other benign tumors.Multimodal imaging—including contrast echocardiography, cardiac MRI, and coronary angiography—should be incorporated into the diagnostic pathway for atrial masses as it may provide valuable preoperative clues and reduce misdiagnosis.

Surgical excision remains the cornerstone of both diagnosis and treatment, typically yielding excellent early outcomes and a favorable overall prognosis. A deeper understanding of optimal management and long-term evolution will require more systematic diagnostic approaches and the coordinated accumulation of cases through multicenter registries.

## Figures and Tables

**Figure 1 life-15-01816-f001:**
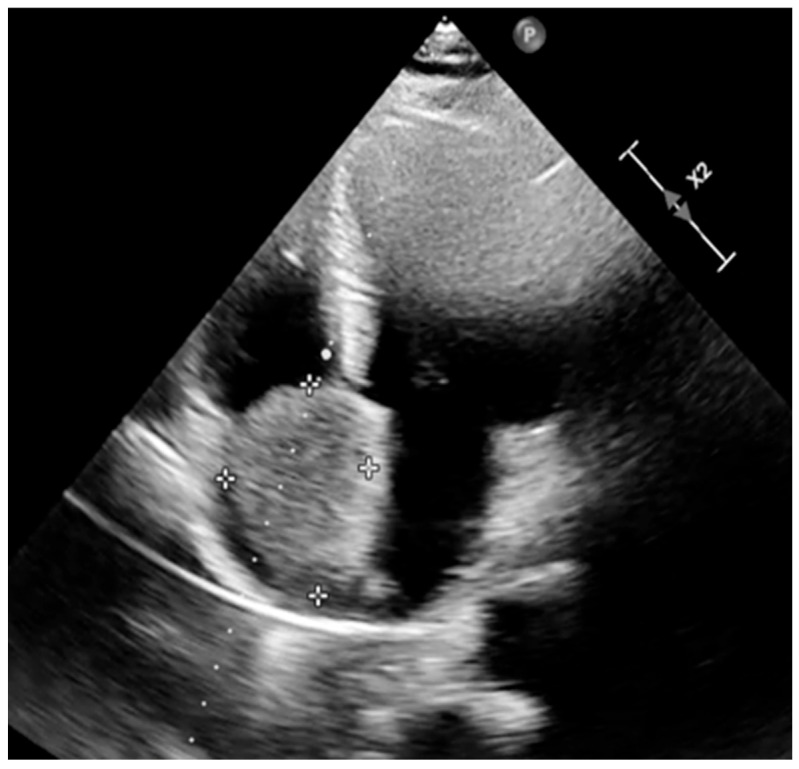
Preoperative transthoracic echocardiography (apical four-chamber view) demonstrating a well-defined intracardiac mass in the right atrium. The mass appears as a homogenous, hyperechoic structure, partially occupying the atrial cavity and exerting a space-occupying effect without causing significant obstruction of atrioventricular inflow at this stage. Tumor margins are clearly demarcated, and no associated pericardial effusion or chamber dilation is evident. These echocardiographic findings raised the suspicion of a benign cardiac tumor, later confirmed histopathologically as a hemangioma. The dotted markers indicate the tumor margins.

**Figure 2 life-15-01816-f002:**
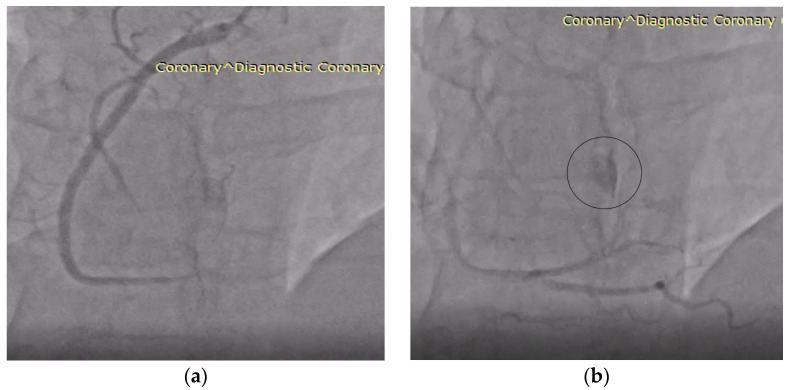
Coronary angiography findings. (**a**) Right coronary artery without angiographic lesions, showing normal course and caliber. (**b**) The same artery shows delayed contrast opacification (“tumor blush”) in the circled area, suggesting either a coronary fistula or a vascular tumor; findings later proved consistent with a cardiac hemangioma.

**Figure 3 life-15-01816-f003:**
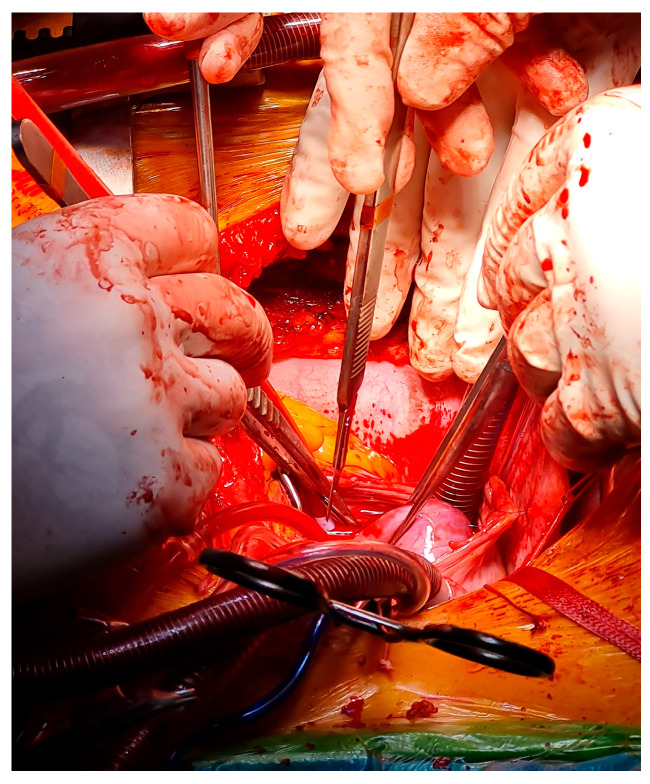
Intraoperative view during open-heart surgery showing the exposure and dissection of the intracardiac tumoral mass consistent with a cardiac hemangioma.

**Figure 4 life-15-01816-f004:**
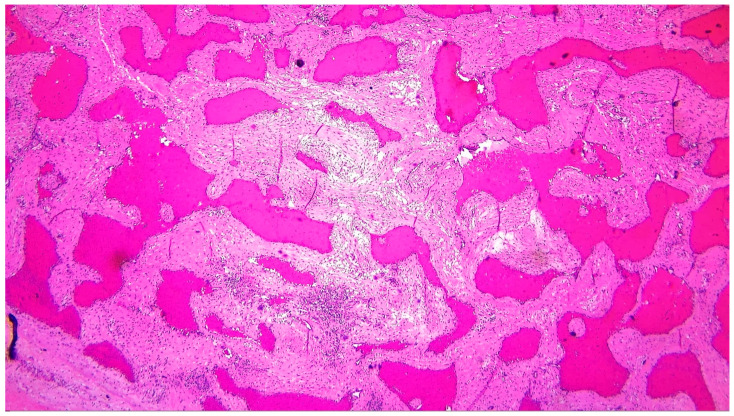
Histopathological examination of the resected cardiac mass (hematoxylin–eosin stain, original magnification ×100) demonstrating large, dilated vascular channels lined by flattened endothelial cells and separated by thin fibrous septa. These features are consistent with a diagnosis of cavernous hemangioma.

**Figure 5 life-15-01816-f005:**
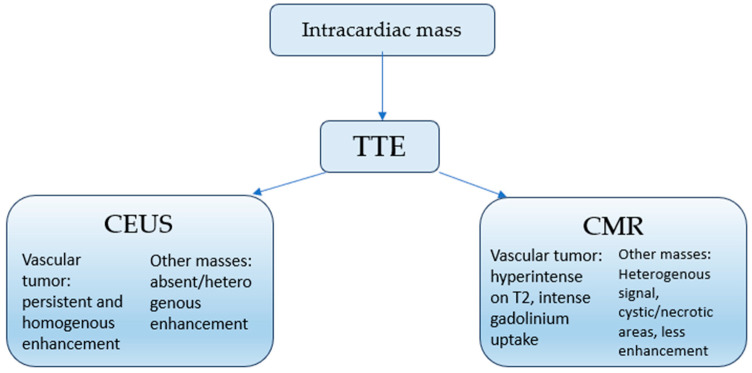
Proposed conceptual framework for multimodal assessment of atrial masses. Coronary angiography may be added to assess tumor perfusion and to exclude coronary fistulas, particularly when surgical planning is anticipated.

**Table 1 life-15-01816-t001:** Clinical timeline of the patient. This table summarizes the key clinical events, investigations, interventions, and outcomes in chronological order.

Date/Interval	Clinical Events and Findings
2021	Total thyroidectomy for benign thyroid disease; levothyroxine replacement initiated.
Months prior to admission	Gradual onset of exertional dyspnea and fatigue, progressing to NYHA class II.
Admission (2025)	Physical examination: afebrile, hemodynamically stable, systolic murmur at the lower left sternal border. Electrocardiography: sinus tachycardia (102 bpm) with left axis deviation. Chest radiography unremarkable. Laboratory tests within normal range.
Day 1 (hospital stay)	Transthoracic echocardiography: 5.5 × 3.5 cm right atrial mass attached at the fossa ovalis; functional tricuspid stenosis with moderate regurgitation; mild degenerative mitral regurgitation; no pericardial effusion.
Day 2 (hospital stay)	Coronary angiography: right-dominant system without stenoses; incidental fistulous connection draining into the right atrium, raising suspicion of a hypervascular lesion.
Surgery (same admission)	Median sternotomy with cardiopulmonary bypass; en bloc excision of the right atrial mass and affected septal tissue; pericardial patch repair of atrial septal defect.
Histopathology	Gross specimen: nodular, spongy mass (5.5 × 5 × 3 cm). Microscopy: cavernous hemangioma characterized by dilated vascular channels lined by flattened endothelial cells, with fibrous and focal myxoid stroma.
Early postoperative course	Transthoracic echocardiography: reduction of tricuspid regurgitation, preserved biventricular function. No residual mass.
Discharge (Day 7)	Patient clinically stable, hemodynamically normal, without pericardial effusion. Discharged on antiplatelet therapy, levothyroxine replacement, and lifestyle recommendations.
Follow-up (1, 3, 6, 12 months, then annually)	Sustained clinical improvement, no recurrence or residual mass on serial echocardiography. Excellent functional recovery.

## Data Availability

The original contributions presented in this study are included in the article. Further inquiries can be directed to the corresponding authors.

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
