# Peer review of "Cardiac Hemangioma in the Right Atrium: Diagnostic Challenges, Imaging Clues, and a Novel Algorithm for Differential Diagnosis"

_life, 2025, doi:10.3390/life15121816_

Round 1

Reviewer 1 Report

Comments and Suggestions for Authors

This work presents a clinically relevant case report of a 47-year-old woman with a rare right atrial cavernous hemangioma, initially misdiagnosed as a myxoma. The authors have highlighted key diagnostic challenges related to cavernous hemangioma and have proposed a structured algorithm that offers improved detection. The manuscript demonstrates solid scientific and methodological rigor for a clinical case report. Considering the limitation of a single case, although the study reflects careful execution and strong rigor, the generalizability of the findings is necessarily limited. In addition, the scientific audience would also benefit from a quantitative comparison of the diagnostic algorithm with [any] existing diagnostic protocols.

Strengths:

  • The novelty of the study is commendable, with a first of its kind evaluation of fMRI and cognitive testing in Ochoa syndrome
  • Strong methodological transparency and analytical depth
  • Well-structured and presented with clarity and logical flow.

Weaknesses:

  • The reporting could benefit from a quantitative analysis of the diagnostic algorithm, to strengthen the interpretability.
  • This can also shed some light on the limitations of the proposed algorithm.
  • Although briefly mentioned, I’d encourage the authors to explicitly note any imaging constraints and generalizability issues.

Author Response

We thank Reviewer 1 for the positive assessment and constructive comments. All requested clarifications have been incorporated into the manuscript and are highlighted in yellow.

  1. Quantitative comparison of the diagnostic algorithm

Response:
Given the exceptional rarity of cardiac hemangioma and the reliance on isolated case reports and small series, no quantitative comparison or validation of diagnostic algorithms is currently feasible. The Discussion now explicitly states that our framework is conceptual rather than guideline-level.

  1. Limitations of the proposed algorithm

Response:
The Limitations section has been revised to clearly acknowledge the rarity of this tumor, the limited number of published cases, and the inability to perform a quantitative assessment. We emphasize that the algorithm is exploratory and requires validation in larger multicenter cohorts.

  1. Imaging constraints and generalizability

Response:
We expanded the Discussion and Limitations to explicitly describe the restricted availability of cardiac MRI and contrast-enhanced echocardiography, and to note the absence of validated imaging criteria for cardiac hemangioma. These factors, together with the scarcity of cases, limit the generalizability of our observations.

Reviewer 2 Report

Comments and Suggestions for Authors

I have read with great attention the manuscript entitled “Cardiac Hemangioma in the Right Atrium: Diagnostic Challenges, Imaging Clues, and a Novel Algorithm for Differential Diagnosis.”
The topic is certainly rare and potentially interesting, as right atrial hemangiomas are exceptionally uncommon, and a detailed case report could contribute to clinical awareness.
However, I have several major concerns that should be addressed before considering the paper for publication.

Major Comments:

  1. The manuscript requires careful English language editing to improve readability, grammar, and overall clarity. The text is verbose and occasionally repetitive, which diminishes its scientific impact.

  2. Proposing a diagnostic algorithm based on a single clinical case appears overly optimistic. While such a proposal may be valuable as a conceptual framework, the authors should substantially tone down their claims and clarify that their algorithm is hypothetical and not validated.

  3. The angiographic and echocardiographic images provided are of suboptimal quality and do not clearly demonstrate the described findings. The authors should improve image quality, resolution, and labeling, or consider reducing the number of non-informative figures.

  4. The abstract does not flow logically from the case presentation to the conclusions. It should be rewritten to better reflect the actual content of the manuscript, emphasizing the rarity of the condition and summarizing key learning points.

  5. The discussion tends to overemphasize the novelty of the case and the proposed diagnostic approach. Given the single-patient nature of the report, this section should instead focus on a concise literature comparison and highlight current diagnostic limitations rather than proposing new clinical guidelines.

Minor Comments:

  • Please verify all references for accuracy and consistency in format.

  • Ensure all abbreviations are defined at first use.

Author Response

We thank Reviewer 2 for the detailed and constructive feedback. All requested revisions have been implemented and are highlighted in pink in the revised manuscript.

  1. English language and clarity

We appreciate this comment. The manuscript has undergone extensive language editing to improve clarity, reduce verbosity, and eliminate repetitive phrasing. The Abstract, Introduction, Discussion, and Conclusions have been streamlined accordingly.

  1. Conceptual nature of the diagnostic algorithm

We agree that a diagnostic algorithm based on a single case should not be presented as validated. The manuscript has been revised to clarify that our framework is conceptual, exploratory, and based on recurrent imaging features described in the literature, including those identified in our recent systematic review. These clarifications have been added to the Discussion and Limitations.

  1. Image quality and figure presentation

We thank the reviewer for this important point. The most informative echocardiographic and angiographic frames have been selected, annotation and labeling have been improved, and non-contributory images have been removed. Imaging constraints are now also acknowledged in the Limitations.

  1. Abstract structure and flow

The Abstract has been rewritten for improved coherence, better alignment with the manuscript content, and clearer emphasis on key diagnostic insights and learning points.

  1. Emphasis in the Discussion

The Discussion has been revised to avoid overstating novelty and to focus more clearly on existing literature, diagnostic challenges, and current limitations. We now explicitly state that the proposed framework is not prescriptive and lacks prospective validation.

Minor comments

  • References: All references have been checked and standardized for accuracy and formatting.
  • Abbreviations: All abbreviations are now defined at first use, and consistency has been ensured throughout the manuscript.

Reviewer 3 Report

Comments and Suggestions for Authors

Areas to Improve:

  • Abstract could be tightened by reducing redundancy (“rarity” and “diagnostic challenges” are mentioned multiple times).
  • Case presentation: could benefit from a subheading for “Investigations and Imaging Findings” to improve readability.
  • References:Standardize citation style and correct minor format issues
Comments on the Quality of English Language

Minor language edits:

  • Change “Our case highlights…” repetition across sections
  • Standardize spacing and punctuation (some missing spaces before parentheses or semicolons).

Author Response

We thank Reviewer 3 for the constructive comments and helpful suggestions. All requested revisions have been implemented and are highlighted in green in the revised manuscript.

  1. Abstract redundancy

Thank you for this observation. The Abstract has been revised to remove repeated references to the rarity of cardiac hemangiomas and to streamline the description of diagnostic challenges. The text now flows more concisely and avoids redundancy.

  1. Structure of the Case Presentation

We appreciate the suggestion. To improve readability, the Case Presentation has been reorganized through clearer paragraph structuring and the addition of brief transition sentences, without modifying the clinical content. These adjustments enhance clarity without introducing a new subheading.

  1. Reference formatting

All references have been reviewed and standardized according to the journal’s requirements. Minor inconsistencies in punctuation, ordering, and formatting have been corrected across the manuscript.

  1. Language consistency

We thank the reviewer for these helpful notes. Repetitive phrasing—particularly the recurrence of “Our case highlights…”—has been removed or rephrased for stylistic variation. Spacing and punctuation inconsistencies have also been corrected throughout the text.

Reviewer 4 Report

Comments and Suggestions for Authors

The case report is interesting.

The histology of hemangioma is well described by the authors. The histology of myxoma can also briefly described in the discussion.

In the limitation section, the authors note that they have not used cardiac MRI or contrast-enhanced echocardiography, although the algorithm they proposed calls for the use of these methods. The discussion should highlight the specificity and sensitivity of these methods for cardiac hemangioma.

Author Response

We thank Reviewer 4 for the positive evaluation and constructive suggestions. All corresponding revisions have been implemented and are highlighted in blue in the revised manuscript.

  1. Histological description of myxoma

Thank you for this helpful suggestion. A concise description of the histopathological features of cardiac myxoma has been added to the Discussion to better contextualize the differential diagnosis and to contrast it more clearly with cavernous hemangioma.

  1. Use of MRI and CEUS in the proposed algorithm

We appreciate the reviewer’s comment. The Limitations section has been expanded to clarify that cardiac MRI and contrast-enhanced echocardiography were not available for the present case. Their inclusion in the algorithm reflects imaging findings reported in previous cases rather than investigations performed in this patient. This distinction is now explicitly stated.

  1. Sensitivity and specificity of imaging modalities

Thank you for this insightful remark. The Discussion has been revised to address the diagnostic performance of cardiac MRI and CEUS. As no validated sensitivity or specificity data exist for cardiac hemangioma, we now state this explicitly and summarize the characteristic imaging patterns consistently reported in the literature (e.g., persistent CEUS enhancement, strong late gadolinium enhancement on MRI) that may support diagnostic differentiation.

Round 2

Reviewer 2 Report

Comments and Suggestions for Authors

Despite the huge amount of work performed by the Authors, I substantially confirm all my previous comments